# Valproic Acid and Its Amidic Derivatives as New Antivirals against Alphaherpesviruses

**DOI:** 10.3390/v12121356

**Published:** 2020-11-26

**Authors:** Sabina Andreu, Inés Ripa, Raquel Bello-Morales, José Antonio López-Guerrero

**Affiliations:** 1Departamento de Biología Molecular, Universidad Autónoma de Madrid, Cantoblanco, 28049 Madrid, Spain; ines.ripa@cbm.csic.es (I.R.); raquel.bello-morales@uam.es (R.B.-M.); ja.lopez@uam.es (J.A.L.-G.); 2Centro de Biología Molecular Severo Ochoa, Spanish National Research Council—Universidad Autónoma de Madrid (CSIC-UAM), Cantoblanco, 28049 Madrid, Spain

**Keywords:** alphaherpesvirus, valproic acid, valpromide, valnoctamide

## Abstract

Herpes simplex viruses (HSVs) are neurotropic viruses with broad host range whose infections cause considerable health problems in both animals and humans. In fact, 67% of the global population under the age of 50 are infected with HSV-1 and 13% have clinically recurrent HSV-2 infections. The most prescribed antiherpetics are nucleoside analogues such as acyclovir, but the emergence of mutants resistant to these drugs and the lack of available vaccines against human HSVs has led to an imminent need for new antivirals. Valproic acid (VPA) is a branched short-chain fatty acid clinically used as a broad-spectrum antiepileptic drug in the treatment of neurological disorders, which has shown promising antiviral activity against some herpesviruses. Moreover, its amidic derivatives valpromide and valnoctamide also share this antiherpetic activity. This review summarizes the current research on the use of VPA and its amidic derivatives as alternatives to traditional antiherpetics in the fight against HSV infections.

## 1. Introduction

The *Alphaherpesvirinae* are a large subfamily of neurotropic DNA viruses that cause epithelial lesions and may subsequently establish latency in sensory ganglia or neurons of the peripheral nervous system, potentially reactivating in the future [1,2]. Members of this group present a broad host range, short replication cycles and the capacity to rapidly spread and destroy the infected cell [3,4]. This subfamily includes a variety of species whose infections cause global health problems in both humans and animals. The alphaherpesviruses that can infect humans are herpes simplex virus type 1 (HSV-1 or HHV-1), herpes simplex virus type 2 (HSV-2 or HHV-2) and varicella-zoster virus (VZV or HVV-3) [5]. HSV-1 is characterized by orofacial (cold sores) and ocular lesions, encephalitis and genital lesions in various age groups [6,7]. However, the most common cause of genital herpes is HSV-2, which is sexually transmitted and also causes recurrent aseptic meningitis [8,9]. On the other hand, VZV causes chickenpox as a primary infection usually in children and then establishes latency, but its reactivation in adults or immunosuppressed individuals can lead to herpes zoster (shingles), which may derive postherpetic neuralgia [10] or lead to aseptic meningitis [9].

In terms of animal health, more than 30 alphaherpesviruses infecting different species (mammals, birds, reptiles, amphibians, and even invertebrates) have been reported [1]. Pseudorabies virus (suid herpesvirus 1, SuHV-1), equid alphaherpesvirus 1 (EHV-1) and bovine herpesvirus 1 and 5 (BoHV-1 and BoHV-5) should be highlighted for causing respiratory problems, neurological disorders and abortions mainly in swine, horses and cattle, respectively, causing significant economic losses in the affected herds. [11,3,12].

According to the WHO, 67% of the global population under the age of 50 are infected with HSV-1 and 13% have clinically recurrent HSV-2. In the case of VZV, its seroprevalence among European children under the age of 5 years reaches 70% or more [13]. Moreover, BoHV-1 affects cattle from many developed and developing countries; the prevalence of the virus in the UK has risen to 83% among unvaccinated herds [12].

Three classes of antivirals are approved for treatment of alphaherpesvirus infections, all of which target viral genome replication: acyclic guanosine analogues; acyclic nucleotide analogues; and pyrophosphate analogues [14]. The most frequently used antiherpetic is acyclovir (ACV), a guanosine analogue which is selectively phosphorylated by the viral enzyme thymidine kinase (TK), and subsequently by other kinases of the infected cell. ACV triphosphate is incorporated by the viral DNA polymerase into the replicating DNA chain, where it inhibits elongation due to the lack of a second hydroxyl group in its structure, thus avoiding the addition of another residue [15,16]. Many nucleoside analogues have been developed which have activity similar to ACV but with enhanced characteristics that improve their pharmacokinetics and pharmacodynamics. Examples of these are oral prodrugs valacyclovir (prodrug of ACV), famciclovir (prodrug of penciclovir) and valgancyclovir (prodrug of ganciclovir), which are effective at lower doses and have greater oral bioavailability compared to ACV [16,17,18].

Unfortunately, the emergence of mutant viruses that are resistant to traditional antiherpetics (especially in immunocompromised patients) and the lack of available vaccines against human HSVs [19] has led to an imminent need for new antivirals. In one study, 4–7% of patients coinfected with HIV and HSV-1 had virus strains which were resistant to approved antiherpetic drugs [20]. In 95% of cases, ACV resistance is acquired via mutations in the TK gene; thus, an alternative treatment strategy is to stop viral replication by targeting the viral DNA polymerase (for example by using the pyrophosphate analogue foscarnet) [21]. Nonetheless, resistance mutations can also occur in the viral DNA polymerase (5% of cases), decreasing its affinity for the nucleoside analogue [22,23,24]. Additionally, none of the antivirals discovered so far protect against the reactivation of herpesvirus in the event that the virus has established latency [2]. Thus, the therapeutic potential of several new drugs is being evaluated, which may be added to potential multi-drug treatment regimens against HSV infections. The current search for new antiherpetics is looking for selective compounds with improved efficacy and minimal adverse reactions, which prevent emergence of resistant mutants by targeting different steps in the viral cycle (adhesion and entry, replication, protein synthesis, assembly or exit). Novel nucleoside analogues as well as specific inhibitors of viral DNA polymerase, viral helicase-primase or ribonucleotide reductase are currently being tested [23,25].

## 2. Valproic Acid

An interesting proposal to reduce the appearance of resistance mutants is to search for compounds which target cellular processes that are essential for the viral life cycle. Valproic acid (VPA; 2-n-propylpentanoic acid or n-dipropylacetic acid) is a branched short-chain fatty acid (Figure 1) clinically used as a broad-spectrum antiepileptic drug in the treatment of neurological disorders including epilepsy, bipolar disorders, neuropathic pain, migraine, and absence, myoclonic, generalized and partial seizures [26,27]. It is also prescribed in the treatment of attention deficit hyperactivity disorder in children, diskynesia, and other disorders that affect learning, thinking and understanding [28]. VPA belongs to the first generation of antiepileptic drugs (AED), which also includes carbamazepine, ethosuximide, phenobarbital, phenytoin and primidone [29,30]. The origin of VPA dates from a hundred years ago, when it was synthesized as a derivate of valeric acid [31], and despite its old age, in 2010 it was the most highly prescribed AED when considering mono- or polytherapeutic uses [28].

The precise mechanism of action of VPA at a molecular level is not fully understood, though it acts at different levels. VPA potentiates the inhibitory activity of the neurotransmitter gamma amino butyrate (GABA) in the central nervous system, by inhibition of GABA degradation and turnover, and by stimulation of its synthesis [32]. VPA also inhibits NDMA receptor-mediated excitatory transmission [33] and reduces the rapid firing of neurons by destabilizing voltage-gated sensitive Na^+^ channels [34,35] as well as T-type Ca^2+^ channels and K^+^ channels. It alters lipid metabolism, downregulates the expression of proteins involved in chromatin maintenance [32], inhibits glycogen synthase kinase 3 (GSK3) and protein kinase A (PKA) [36]. Inhibition of histone deacetylase (HDAC) by VPA makes it an interesting candidate for cancer and HIV treatments [37]. In fact, VPA plays an important role in new virotherapies for cancer, based on the administration of oncolytic herpesvirus vectors combined with drug therapy to enhance tumor eradication [38].

Previous studies have positioned VPA as a potential broad-spectrum antiviral, since it inhibited in vitro infection by enveloped viruses from various viral families, though it showed no effect against non-enveloped viruses [39]. VPA caused a drastic reduction of the yield of the flaviviruses West Nile virus (WNV) and Usutu virus (USUV), the arenavirus lymphocytic choriomeningitis virus (LCMV), the rhabdovirus vesicular stomatitis virus (VSV), the togaviruses Semliki forest virus (SFV) and Sindbis virus (SINV), the asfivirus African swine fever virus (ASFV) and the poxvirus vaccinia virus (VACV). On the contrary, VPA had no effect on the nonenveloped viruses tested, including foot-and-mouth disease virus (FMDV), encephalomyocarditis virus (EMCV), bovine enterovirus (BEV), and equine rhinitis A virus (ERAV) (picornaviruses). The results suggested that VPA can act at different steps of enveloped virus infection; although WNV RNA and protein synthesis were drastically abolished, VSV RNA and protein synthesis remained unaltered [39]. Considering that alphaherpesviruses present a lipid bilayer membrane, it was hypothesized that VPA would also have an antiviral effect against this viral subfamily.

In fact, the antiviral activity of VPA against HSV-1 has been demonstrated, in our group, in the human oligodendrocyte cell line HOG [40]. This cell line is a suitable cell model for the investigation of demyelinating disorders such as multiple sclerosis, an autoimmune disease whose etiology may involve some herpesviruses [41,42,43,44]. In that study, the half maximal inhibitory concentration (IC_50_) for VPA inhibition of HSV-1 infection was 0.55 mM [40], comparable to the regular dosage recommended for treatment of epilepsy which ranges between 0.3 to 0.6 mM in the plasma [45]. Moreover, cytotoxicity assays showed that HOG cells incubated with 4 mM of VPA for 48 h maintained more than 80% of viability [40]. Although clinical trials in humans have not yet been carried out, its AED dosage may be sufficient for antiviral activity against HSV-1. This is supported by the results of a nested case-control study, in which patients on VPA therapy lasting longer than 90 days were associated with a downward trend in risk of herpesvirus infection [26].

Nowadays, the use of VPA is limited and under surveillance due to its teratogenic and hepatotoxic effects. Indeed, as VPA can alter the gene expression of approximately 20% of the transcriptome in a tissue-dependent manner [32], predicting the effects of this drug can be a real challenge. The most important described teratogenic effects for VPA are major congenital malformations and neural tube defects in embryo and fetus, and thus is not recommended for women of childbearing age [46,47]. This teratogenicity is achieved through its structural characteristics: having a carboxylic acid, branching in carbon atom 2 (C-2) with two side chains, each containing at least three carbon atoms, and presenting a hydrogen atom at C-2 [48] Regarding hepatotoxicity, it is suggested that VPA metabolism leads to two hepatotoxic metabolites (4-ene-VPA and 2,4-diene-VPA), which may form an intermediate thioester with the free form of the mitochondrial coenzyme A in the liver [49,50]. Another hypothesis proposed that the hepatotoxicity associated with VPA originates from the inhibition of pyruvate transport, ATP- and GTP-dependent succinate-CoA ligases, hepatic N-acetylglutamate synthase and α-lipoamide dehydrogenase [51].

## 3. Amidic Derivatives of Valproic Acid

With the intention of reducing the toxic effects associated with VPA, some of its amidic derivatives have been tested as potential antivirals against alphaherpesviruses, the so-called “second-generation” VPAs. Valpromide (VPD; dipropylacetamide or 2-propylvaleramide) and valnoctamide (VCD, valmethamide or 2-etyhl-3-methyl-pentanamide) are aliphatic amide derivatives of VPA that barely differ in structure, containing an amide group (see Figure 1). This slight difference in the acidic core molecule (absence of a free carboxylic group) might be responsible for the lack of hepatotoxic and teratogenic effects [52,53,54]. Both drugs have been commercialized in various European countries for more than 50 years [54].

### 3.1. Valpromide

VPD is the primary amide of VPA and is currently used as an anticonvulsant and antipsychotic drug authorized in all EU Member States. It is indicated for treatment of primary generalized epilepsies and bipolar disorder in cases when lithium therapy is not tolerated [55,56]. VPD acts as a prodrug which, after oral or intravenous administration, is rapidly biotransformed through metabolic hydrolysis into its corresponding acid (90% of the total VPD dose) [54,57]. VPD easily crosses the blood-brain barrier and seems to have an anticonvulsive effect 3 to 5 times more potent than that of VPA [48,58]. In terms of teratogenicity, VPD does not show any such effect in pregnant rats, mice, or dogs [59,60]. Nevertheless, since VPD is transformed into VPA in humans, this drug is still contraindicated in women of childbearing age or pregnant women, since it may exhibit the same teratogenic effects [54,61].

The antiviral effect of VPD against HSV-1 was tested in our laboratory in human and murine cell lines and primary cultures based on rat oligodendrocyte precursor cells (OPCs), revealing an antiherpetic effect comparable to ACV. VPD was tested at 0.5 mM, a concentration compatible with the therapeutic doses already approved in humans. Indeed, OPCs viability in the presence of 0.5 mM VPD during 24 h was close to 100% [49]. These promising results encourage testing VPD antiviral activity in animal models and possibly in human clinical trials. Although VPD in humans is biotransformed into teratogenic VPA and therefore would not be an ideal drug in the treatment of infections in pregnant women, it could be used in the treatment of animal species where VPD does not carry such metabolization.

### 3.2. Valnoctamide

VCD is the primary amide of valnoctic acid and a constitutional isomer of VPD. Actually, it is a racemic compound which encompasses four stereoisomers, all of them showing a higher anticonvulsant efficacy in rodent models of epilepsy in comparison to VPA (about 3 to 10 times higher) [62,63]. In particular, the stereoisomer (2S,3R)-VCD stands out for having a greater oral clearance and a shorter half-life [52,64]. VCD is used as an anxiolytic and it has been involved in various clinical trials for acute mania and bipolar disorder [65,66]. Unlike VPD, VCD does not biotransform into VPA, so its pharmacological activity is different and its half-life is ten times longer than that of VPD [55]. Moreover, VCD has shown no teratogenic effects in murine models [67,68].

VCD’s antiviral effect against HSV-1 was also tested in HOG cells at the concentration 0.1 mM (VCD shows no cytotoxicity in this cells at this concentration), and it significantly inhibited the infection at accepted human dosage, similar to the effect achieved with ACV [49].

### 3.3. Antiviral Mechanism of VPA and Related Derivatives

Little is still known about the antiviral mechanism of VPA and its amidic derivatives. The fact that VPA acts at different molecular levels gives this compound the characteristic of broad-spectrum antiherpetic. This may be explained by an interference with different cellular and viral processes. Some studies have revealed that VPA and VPD may interfere with the initial steps of the viral cycle, specifically during virus entry [49]. As mentioned previously, VPA affects lipid metabolism (including the synthesis of phosphatidylinositols) and formation of cell membranes, so it may alter the viral acquisition of lipid envelope, resulting in mature viral particles with low stability [45]. Furthermore, some studies have reported antiviral activity of VPA, VCD and VPD against cytomegalovirus (CMV), a virus belonging to the *Herpesviridae* family that causes brain defects in neonates, not only outside the central nervous system but also in the developing brain of rodents [65,69]. CMV attaches to the cell through interaction of its glycoprotein gB with cellular heparan sulfate proteoglycans, and both VCD and VPD appear to affect the recognition of gB [49,69]. This mechanism may also explain the antiviral activity of these drugs against HSV-1. Moreover, some studies demonstrated that VPA inhibits the reactivation of Epstein-Barr virus (EBV) lytic cycle in vitro due to VPA’s HDAC inhibitory activity. However, despite the fact that VPD does not have this HDAC inhibitory property, it still inhibits reactivation of the EBV lytic cycle [70,71]. Furthermore, a recent study has shown that VPA could also have a potential antiviral effect against RNA virus SARS-CoV-2, as it reduces the number of angiotensin-converting enzyme 2 (ACE-2) in endothelial cells, the main cellular receptor that this coronavirus uses to enter the cell [72]. Thus, further research needs to be performed to unravel the antiviral mechanism of VPA and related derivatives.

## 4. Conclusions

There is proof that VPA and its amidic derivatives have shown antiviral activity against various members from the *Herpesviridae* family, specifically against alphaherpesviruses that can infect humans. Nowadays, the high incidence of alphaherpesvirus infection, especially HSVs, must be taken as a major concern, and the need for effective therapies also in animals should not be forgotten. The fact that VPA acts at different molecular levels gives this compound the characteristics of a broad-spectrum antiherpetic. Hence, it would be very interesting to test the potential inhibitory activity of VPA and its amidic derivatives against animal infections due to the importance of some alphaherpesviruses such as SuHV-1, BoHV-1 or EHV. Considering that VPA, VPD and VCD are already licensed for clinical use by the EMA (European Medicines Agency), the time required to approve their brand-new use for the treatment of alphaherpesvirus infections should be shorter than for novel drugs. In this way, VPA and its amidic derivatives VPD and VCD may be part of the solution to the growing problem of viral resistance against traditional antiherpetics.

## Figures and Tables

**Figure 1 viruses-12-01356-f001:**
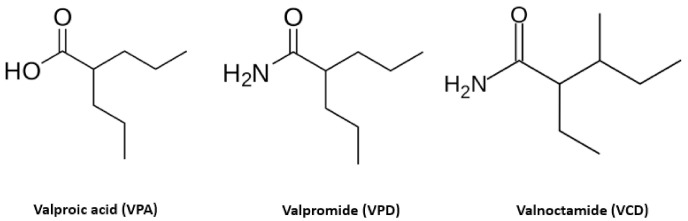
Molecular structure of valproic acid (VPA) and its amidic derivatives valpromide (VPD) and valnoctamide (VCD). VPD and VCD differ from VPA in structure as they contain an amide group instead of a carboxylic group.

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
