# Peer review of "Valproic Acid and Its Amidic Derivatives as New Antivirals against Alphaherpesviruses"

_viruses, 2020, doi:10.3390/v12121356_

Round 1

Reviewer 1 Report

Manuscript written by Andreu S and others  authors describes the data of anti-HSV activity of Valproic acid and its amidic derivatives . Herpes Simplex Viruses particular HSV-1 and HSV-2 present a global health problems in animals and humans . Current anti- HSV therapy based on the use of nucleoside analogues such as acyclovir is successful but  the emergence of ACV resistant mutants and absence of effective anti-HSV vaccine requires development of novel anti herpetic drugs.Authors  described the last experimental data about anti-HSV activity of VPA and others amidic derivatives such as valpromide and valnoctamide  in cell culture  of HOG cells that represent a suitable model for study of demyelinating disorders. IC50 for VPA is 0.5 mM, it is a quite high dose for such drugs. Authors indicated the major problem  for therapeuitic use of VPA is toxicity in humans, particular the teratogenic and hepatogenic effects. They didn't not provide toxicity data on HOQ cells. For others  tested anti-viral compounds such as VPD and VCD authors also didn't provide toxicity data on HOQ and others cells that make difficult to establish the most important value for human therapy such as therapeutic index  (TD50/EC50).   That jeopardise their conclusion about approval  tested drugs as anti HSV compounds. 

Author Response

Thank you for reviewing my article. Here I present the points that I have added about cytotoxicity data (Please see the attachment).

Point 1. Toxicity data of valproic acid (VPA) in HOG cells:

I added the following information in line 119. "Moreover, cytotoxicity assays showed that HOG cells incubated with 4 mM of VPA for 48 h maintained more than 80% of viability [40]".

Point 2. Toxicity data of (valpromide) VPD and (valnoctamide) VCD in HOG cells:

VPD: I added the following information in line 164: "Indeed, OPCs viability in the presence of 0.5 mM VPD during 24 h was close to 100% [49]".

VCD: In the case of VCD, i added this sentence in line 179: "VCD’s antiviral effect against HSV-1 was also tested in HOG cells at the concentration 0.1 mM (VCD shows no cytotoxicity in this cells at this concentration)..."

Reviewer 2 Report

This is a well written review describing valproic acid and derivatives and their effect on HSV replication. The review presents an interesting case that these compounds deserve increased attention in both animal and human studies as a antiherpetic treatment.

This review addresses the use of valproic acid and its derivatives as an anti-herpetic drugs. It argues that these compounds should be considered as potential treatments. The authors provide both pros and cons to the use of these drugs including several reports demonstrating their effect on viral replication (HSV-1) and at least one report on viral infections in patients being treated with these compounds. A strength of the review is that the authors also point out potential cons for their use including hepatic toxicity and fetal abnormalities. The authors make a nice case for additional studies both in vitro and in vivo targeting valproic acid derivatives in treating and herpesvirus infections and possibly preventing viral reactivation. Overall the review is well written and referenced.

Author Response

I would like to thank this referee for the review of this article.